# Physicochemical Changes of Heat-Treated Corn Grain Used in Ruminant Nutrition

**DOI:** 10.3390/ani12172234

**Published:** 2022-08-30

**Authors:** Bojana Kokić, Ljubica Dokić, Lato Pezo, Rade Jovanović, Nedeljka Spasevski, Jovana Kojić, Miroslav Hadnađev

**Affiliations:** 1Institute of Food Technology, University of Novi Sad, Bulevar cara Lazara 1, 21000 Novi Sad, Serbia; 2Faculty of Technology, University of Novi Sad, Bulevar cara Lazara 1, 21000 Novi Sad, Serbia; 3Institute of General and Physical Chemistry, University of Belgrade, Studentski Trg 12-16, 11000 Belgrade, Serbia; 4Institute for Science Application in Agriculture, Bulevar despota Stefana 68b, 11000 Belgrade, Serbia

**Keywords:** corn, starch, ruminant, heat treatment

## Abstract

**Simple Summary:**

Whole grain with an intact pericarp is highly resistant to digestion by ruminants and external processing is required. Cereal grain is processed using different combinations of heat, moisture, time, and mechanical action. As a result of processing, starch becomes more available to both ruminal microbial and pancreatic enzymes. In animal science literature, there is a lack of data regarding the physicochemical properties of starchy materials prior to and after the application of processing treatments, as well as the connection to biological responses of animals in nutritional studies. Therefore, the aim of the present study was to quantitatively represent physicochemical properties of raw and heat-treated corn starch using an in vitro methodology that is commonly used by food scientists. The heat treatments examined were pelleting, steam flaking, micronization, and extrusion of corn. The obtained results indicated significant differences in gelatinization, hydration, and pasting properties of corn starch depending on the heat treatment applied. Future research should aim for the determination of rumen starch degradation traits and possible correlations with the present results.

**Abstract:**

Cereal grain is processed using different combinations of heat, moisture, time, and mechanical action in order to improve its digestibility. The objective of the present research was to quantitatively represent the physicochemical properties of raw and processed starch using an in vitro methodology, as well as to describe the changes that occurred after heat treatment, such as pelleting, steam flaking, micronization, and extrusion of corn. Based on the obtained results, pelleting, steam flaking, and micronization can be considered as mild heat treatment methods, whereas extrusion proved to be a severe heat treatment method. Analysis of functional and pasting properties implied a possible interaction between the degraded components in the steam-flaked sample, as well as in the micronized sample, through to a lesser extent. Additionally, the occurrence of dextrins was noted after extrusion. The obtained results indicate the existence of significant differences in the physicochemical properties of corn starch depending on the heat treatment applied, which could possibly affect rumen starch degradation traits.

## 1. Introduction

Cereals differ significantly in rumen fermentation characteristics, largely due to the physical and chemical properties of cereal starch. Starch accounts for 45–75% of cereal dry matter [1] and is important in ruminant nutrition as an energy source, especially in ruminant diets used to promote high levels of production. The digestibility of starch generally depends on the granules’ characteristics such as size, degree of crystallinity, degree of polymerization, non-starch components, their interactions with starch, and their amylose:amylopectin ratio [2]. Additionally, the type and degree of processing are additional factors, and the resulting physicochemical changes affect the structural, molecular, and functional characteristics [3]. Therefore, an examination of the above-mentioned characteristics and their correlation with digestibility is necessary to understand the effects of heat treatments on digestion rate.

According to Huntington [4] whole grain with an intact pericarp is highly resistant to digestion by ruminants due to the inability of rumen bacteria to attach to its surface. Up to 30% of the whole grain can appear in the feces of cattle fed whole grain diets [5]; therefore, external processing is required. Cereal grain is processed using different combinations of heat, moisture, time, and mechanical action. These processing treatments enable bacterial attachment to starch granules, thus increasing its digestibility. Additionally, processing breaks down the kernel endosperm structure, disrupts the protein matrix surrounding the starch granules in the endosperm, and gelatinizes the starch granules [6,7]. As a result of processing, starch becomes more available to both ruminal microbial and pancreatic enzymes [8].

The various processing treatments of animal feeds influence the rumen fermentation characteristics of grain starch, which in turn affect animal performance. Based on a review of the available literature, the most commonly studied treatments of corn used in ruminant nutrition are grinding, dry rolling, steam rolling, and steam flaking [4,7,8,9,10,11,12,13,14,15,16,17,18,19,20,21,22,23,24,25]. Studies covering pelleting [26,27], extrusion [24,28,29,30], and expansion [31] are available but to a much lesser extent than the above-mentioned treatments. On the other hand, a detailed search of available studies revealed a lack of research regarding the nutritive value of micronized corn in ruminant nutrition.

It has come to the authors’ attention that in animal science literature, descriptions of processing methods are very often scarce and incomplete, and relevant processing variables are often omitted. Additionally, in order to correctly interpret the biological responses of animals in nutritional studies, data regarding the physicochemical properties of starchy materials prior to and after the application of processing treatments should be included in the discussion. In most studies in which the influence of certain treatment on cereal starch and its impact on animals is analyzed, there is a lack of data regarding the changes in the starch component, as well as the induced intensity of those changes. Almost half a century ago, Hale [32] emphasized that the gelatinization of the grain starch is indicative of animal utilization, but the notable improvement may not be entirely due to the occurrence of gelatinization. He also stated that using gelatinization of the starch as the only evaluation measurement of processed grains has limitations. However, in spite of this, gelatinization of starch is usually the only parameter used to quantitatively describe the changes of grain starch in animal science literature and unfortunately, this is only in a minority of the research.

According to White et al. [33], reliable in vitro methodologies that are commonly used by food scientists can be used for an accurate and quantitative representation of the physicochemical properties of raw and processed materials. Since variations between and within processing treatments affect the physicochemical properties of starch differently, the choice of an adequate processing treatment is a fundamental consideration, as it can have a marked effect on the properties of starch, as well as on the starch’s digestibility.

The objective of the present research was to quantitatively represent the physicochemical properties of raw and processed starch using an in vitro methodology, as well as to describe the changes that occur after heat treatments such as pelleting, steam flaking, micronization, and extrusion of corn. The goal of this research was set in order to fill in the gap which evidently exists in animal science literature, as well as to gain principal knowledge about the physicochemical properties of micronized, pelleted, and steam- flaked corn starch which is currently scarce and to confirm the existing knowledge from the food science field regarding extrusion. Changes in the structure of starch granules were monitored by scanning electron microscopy (SEM) and differential scanning calorimetry (DSC), while changes in the hydration and swelling of samples were determined by analyzing the pasting properties, water absorption index (WAI), and water solubility index (WSI) of the samples. Principal component analysis (PCA) was carried out to assess the correlation between the different physicochemical properties of unprocessed and heat-treated corn samples.

## 2. Materials and Methods

### 2.1. Material

Corn (*Zea mays*) variety ZP 666 grown in Serbia was obtained from local producer. The initial moisture content of the corn sample was determined to be approximately 10.96%, while the content of starch was 75.63% on a dry matter basis.

### 2.2. Heat Treatments

#### 2.2.1. Steam Flaking

The steam flaking technological process of whole corn grain was carried out at the factory for feed production “Opulent”, Čurug, Serbia, using the equipment for steam flaking of cereals produced by Ottevanger, Aalten, Netherlands. The corn grain was kept in the steam chest above the device where the material was conditioned for 30 min with the addition of 10bar steam pressure. The moisture content of the corn grain after conditioning was 14.36% and a temperature of 85 °C was reached. The final shape of the processed grain was obtained by passing the material between two rollers.

#### 2.2.2. Micronization

The technological process of micronization of whole corn grain was carried out at the factory for feed production “Komponenta”, Ćuprija, Serbia. The corn grain was kept in the steam chest above the device where the material was conditioned for 15 min by adding 4bar steam pressure. The material was dosed in a thin layer on a vibrating line (1.1 × 3.4 m) from the steam chest. The moisture of corn grain after conditioning was 14.29% and the material flow rate was 500 kg/h. The temperature of the corn grain was measured at the beginning, middle, and end of the vibrating line and was 95, 106, and 111 °C, respectively. The final shape of the processed grain was obtained by passing the material between two rollers.

#### 2.2.3. Pelleting

The technological process of pelleting was carried out at the pilot plant of the Institute of Food Technology in Novi Sad, Serbia. Corn grain was first ground to pass through a 4 mm sieve using a hammer mill Type 11, ABC Engineering, Pančevo, Serbia. The mixing and conditioning of the corn was performed on a double-shaft paddle mixer-conditioner, model SLHSJ0.2, Muyang, Yangzhou, China. Steam was added directly into the conditioner until the material reached a temperature of 80 °C, after which it was discharged into the bin under the mixer-conditioner. The moisture of the conditioned material prior to pelleting was 16.52%. The material was pelleted on a flat die pellet press, model 14-175, Amandus Kahl, Reinbeck, Germany. The diameter of the pellets’ die opening was 6 mm and the die thickness was 36 mm. The material flow was 15.4 kg/h and the pelleting temperature was 61 °C.

#### 2.2.4. Extrusion

The technological process of extrusion was carried out at the pilot plant of the Institute of Food Technology in Novi Sad, Serbia. Prior to extrusion, corn grain was milled and conditioned using the same procedure described for pelleting. The moisture content of the conditioned material prior to extrusion was 17.31%. Extrusion processing was done in a simple pilot single-screw extruder, model OEE 8, Amandus Kahl, Reinbeck, Germany. The number of openings on the extrusion die was 2, with a diameter of 8 mm. The feeding rate of material was 19.8 kg/h and the screw speed was 75 Hz. The temperature of the material measured at the end of the extrusion barrel was 95 °C.

### 2.3. Differential Scanning Calorimetry (DSC)

Changes in the crystalline structure of starch during gelatinization were monitored by differential scanning calorimetry using a DSC 204 F1 Phoenix^®^, Netzsch, Erlangen, Germany. The calorimeter was calibrated using indium as a standard. Approximately 4 mg of ground corn sample was measured directly in the DSC pan, after which water was added so that the sample/water ratio was 1:2. The pans were hermetically sealed and left overnight. The samples were heated at a rate of 10 °C/min from 20 to 95 °C. An empty DSC pan was used as a reference. The calorimetric method measured the enthalpy of gelatinization (ΔH) of the sample, i.e., the energy needed to achieve complete starch gelatinization. ΔH was determined from the peak area on the thermogram and expressed in J/g, calculated on a dry matter basis. The onset temperature (T_o_), peak temperature (T_p_), end temperature (T_e_), and gelatinization enthalpy (ΔH) were analyzed and calculated by DSC software Proteus^®^. The measurements were repeated in triplicate.

The degree of gelatinization (DG) was calculated by comparing the ΔH of the unprocessed sample with a heat-treated sample [34]:DG (%) = (1 − ∆H_treated_/∆H_unprocessed_) × 100,(1)

### 2.4. Water Absorption Index (WAI) and Water Solubility Index (WSI)

The water absorption index (WAI) and water solubility index (WSI) were determined and calculated as described by Lazou and Krokida [35]. The measurements were repeated in triplicate.

### 2.5. Paste Consistency (Pasting Profile)

The pasting properties were evaluated using the rheometer HAAKE MARS, Thermo Scientific, Darmstadt, Germany. Pasting profiles of samples were recorded using 15% corn suspensions (calculated on a dry matter basis) with the addition of 60 mL distilled water. Each sample was held at 25 °C for 3 min, followed by heating to 95 °C, after which the temperature was held constant at 95 °C for 15 min before cooling to 25 °C. The heating and cooling rates were set at 1.5 °C/min, while a shear rate of 10 s^−1^ was applied. The recorded parameters were initial viscosity (IV) (at the beginning of the heating phase), peak viscosity (PV) (timepoint at which most of the granules reached maximum swelling), final viscosity (FV), and gelatinization temperature (T_g_) (temperature when the viscosity starts to rise due to the beginning of starch granule swelling). The measurements were repeated in triplicate.

### 2.6. Scanning Electron Microscopy (SEM)

The ground samples were steamed with gold on a Sputter Coater SCD 005, BALTEC SCAN (WD = 50 mm, for 90 s, power 30 mA) and observed by scanning with an electron microscope JMS SEM 6460 LV, JEOL Ltd.,Tokyo, Japan, at an acceleration voltage of 25 KV. Magnification is shown at the bottom of each image.

### 2.7. Statistical Analysis

Experimental data were analyzed by means of an analysis of variance (ANOVA) and reported as mean value ± standard deviation. Significant differences were estimated via Tukey’s HSD test (*p* ≤ 0.05). The correlations between the certain physicochemical properties of starch were evaluated using Pearson’s correlation (significance level *p* ≤ 0.01 or *p* ≤ 0.05). With the aim to evaluate the effects of the different heat treatments on the physicochemical properties of starch, a principal component analysis (PCA) was conducted. Statistical methods were performed using Statistica 12.0 software (Statsoft, Tulsa, OK, USA).

## 3. Results

### 3.1. Gelatinization Properties

In Figure 1, DSC thermograms of the unprocessed and heat-treated corn are shown, and in Table 1, the values obtained from the thermograms are displayed. The endothermic peak shows gelatinization of the starch. The surface between the baseline and the thermogram represents the enthalpy of gelatinization (ΔH). The largest ΔH was determined for the unprocessed corn sample (4.30 J/g), and the smallest gelatinization temperature range (ΔT) was 8.4°C. After the applied heat treatments, a decrease in ΔH in all corn samples when compared to an unprocessed sample was noted, indicating that a part of starch was gelatinized. Additionally, complete gelatinization of starch was recorded in one sample (EC) based on the absence of the gelatinization endotherm. The highest T_o_ was obtained for the unprocessed sample (71.0 °C). The applied heat treatments induced a significant decrease in T_o_ which ranged from 66.9 to 68.7 °C. Consequently, the reduction in T_o_ led to an increase in ΔT, the temperature range of gelatinization of starch granules remaining after heat treatment. The DG of the heat-treated corn was determined based on the change of ΔH compared to the unprocessed corn (Table 1). After applying the heat treatments to the corn, partial to complete gelatinization of starch occurred, ranging from 35.64 to 100%.

### 3.2. Water Absorption Index (WAI) and Water Solubility Index (WSI)

Analysis of the obtained values for the WAI and WSI (Table 1) revealed statistically significant differences between the samples (*p* ≤ 0.05). The lowest WAI was determined to occur in the unprocessed sample (2.77 g/g), in which the starch was in its native form. The crystallinity of starch limits its swelling and therefore the amount of water it can absorb. Gelatinization results in a loss of crystallinity, which causes the granules to swell freely and absorb much more water, as demonstrated by the increase in WAI of all heat-treated samples. Pelleting, steam flaking, and micronization of corn led to a moderate increase in WAI (3.35–3.86 g/g), to about the same extent. Extrusion produced the highest WAI (6.59 g/g), indicating a greater degree of damage to the starch granules.

Pelleting, steam flaking, and micronization of corn led to a numerical decrease in WSI in comparison to the unprocessed sample; however, only the reduction induced by steam flaking was statistically significant (3.37 vs. 6.20 g/100 g, *p* ≤ 0.05). On the other hand, extrusion was the only heat treatment that produced ab increase in WSI (20.89 g/100 g).

### 3.3. Pasting Properties

The pasting profiles of unprocessed and heat-treated corn samples are presented in Figure 2 and the IV, PV, FV, and T_g_ values obtained from the pasting profiles are displayed in Table 1. The heat treatments influenced the occurrence of significant differences in the pasting properties of the analyzed corn samples. All samples showed a gradual increase in viscosity with the heating of the suspension, except for sample EC (Figure 2), indicating that in these samples some of the starch remained ungelatinized after heat treatment, whereas a complete gelatinization of the starch has occurred after extrusion. In the unprocessed sample, the recorded IV was 1.07 Pa·s. After pelleting, steam flaking, and micronization, IV significantly decreased and was in the range of 0.42–0.77 Pa·s. In contrast, after extrusion a noticeable increase in IV was obtained (19.90 Pa·s). The highest values of PV and T_g_ were obtained in the unprocessed sample (68.08 Pa·s and 77.72 °C, respectively). After the applied heat treatments, both parameters significantly decreased and were in ranges of 30.41–53.46 Pa·s and 72.14–73.57 °C, respectively. Changes in FV were also noted for all analyzed samples. The highest FV value was recorded in sample SFC (248.60 Pa·s) and the lowest in sample EC (39.61 Pa·s).

### 3.4. Scanning Electron Microscopy (SEM)

A visual representation of the microstructure of ground corn grain before and after the heat treatments is shown in Figure 3. The native starch granules in sample UC (unprocessed) have a characteristic morphology, i.e., shape and size that correspond to corn starch granules [36]. The heat treatments led to damage of the corn structure to varying degrees depending on the choice and intensity of treatment. The biggest change in the morphology of starch granules was caused by the extrusion treatment. Through comparison of the images of the unprocessed and extruded samples, a complete disintegration of the starch granules caused by extrusion can be observed, followed by the formation of large conglomerates of different shapes with a relatively smooth and surface that is rough in some places. After pelleting, steam flaking, and micronization, part of the starch granules remained intact, i.e., the native structure was not damaged during these treatments, while the remaining granules gelatinized and formed clumps of different sizes.

### 3.5. Principle Component Analysis (PCA)

PCA was applied to visualize the relationship between all the measured variables and to present the results in plots that can be used for simple interpretation. Four treatments (pelleting, steam flaking, micronization, and extrusion) and twelve variables were considered (Figure 4). PC1 and PC2 accounted for 89.97% and 6.82% of the total variance, respectively. PC1 and PC2 together accounted for 96.79%. The properties whose curves lie close to each other on the plot are positively correlated, while those whose curves run in opposite directions are negatively correlated. PCA analysis showed that T_o_, T_p_, T_e_, and T_g_ were clustered and negatively correlated with PC1. On the other hand, WSI and IV were also clustered, but were positively correlated with PC1.

A positive correlation of DG with WAI (r = 0.951, *p* ≤ 0.05) and a negative correlation with PV (r = −0.967, *p* ≤ 0.01) can be observed in the PCA loading plot. WAI showed positive correlations with WSI (r = 0.936, *p* ≤ 0.05) and IV (r = 0.957, *p* ≤ 0.05) and negative correlations with T_o_ (r = −0.971, *p* ≤ 0.01), T_p_ (r = −0.965, *p* ≤ 0.01), T_e_ (r = −0.962, *p* ≤ 0.01), ∆H (r = −0.952, *p* ≤ 0.05), PV (r = −0.955, *p* ≤ 0.05), and T_g_ (r = −0.978, *p* ≤ 0.01). Furthermore, WSI was positively correlated with IV (r = 0.985, *p* ≤ 0.01) and negatively correlated with T_o_ (r = −0.979, *p* ≤ 0.01), T_p_ (r = −0.981, *p* ≤ 0.01), T_e_ (r = −0.984, *p* ≤ 0.01), ∆T (r = −0.959, *p* ≤ 0.01), and T_g_ (r = −0.982, *p* ≤ 0.01). Other correlations can be found in Appendix A.

The score plot (Figure 5), used in the classification of heat treatments, showed that PC1 clearly separated two major groups: corn samples that were unprocessed, pelleted, steam-flaked or micronized were mostly in the negative range, while only the extruded corn sample was in the positive range. Additionally, analysis of PC2 showed that the unprocessed corn sample was in the negative range, distant from all heat-treated samples, whereas the pelleted, steam-flaked and micronized corn samples were in the positive range.

## 4. Discussion

### 4.1. Gelatinization Properties

Since starch gelatinization is an endothermic process, differential scanning calorimetry (DSC), which measures the temperature and enthalpy of gelatinization, is very often used to monitor this process. According to Singh et al. [37], ΔH, an overall measure of crystallinity (quality and quantity), is an indicator of the loss of molecular order within the granule, which was noted in all heat-treated corn samples based on a decrease in ΔH as compared to an unprocessed sample. Additionally, Altay and Gunasekaran [38] state that T_o_ is a measure of the perfection of starch crystallites among all granules present. Therefore, the lower T_o_ observed in the heat-treated corn samples indicates a disorganization of the starch granule crystalline structure, i.e., less perfect crystallites. A consequent increase in ΔT, which was noted in all heat-treated samples, indicates a higher inhomogeneity of the starch granules, which is due to the fact that in the heat-treated samples, certain granules were completely gelatinized, some only partially gelatinized, while one part of the granules remained unchanged [39].

In general, gelatinization depends on the presence of water. Svihus et al. [40] state that when the water content is high, most starches will gelatinize at temperatures between 50 and 70°C, but with limited water contents (below ~40%), the gelatinization temperature will increase. During steam flaking treatment, the corn grain was conditioned for 30 min and a temperature of 85°C was reached. Under the given conditions of steam flaking, a slightly lower DG was determined in sample SFC than in the studies carried out by Medel et al. [41] (42.51 versus 50%). Medel, Latorre, de Blas, Lázaro, and Mateos [41] found that gelatinization occurred in 50% of the starch in corn grain that was conditioned for 50 min at 99°C and then flacked by passing between the rolls. This finding was also confirmed by Qiao et al. [42]. In a study by Han, Guo, Cai, and Yang [24], 67.29% of corn starch gelatinized after steaming (100–110°C) the whole grain for 50 min, after which the moisture level was 21–23%. A smaller DG of sample SFC obtained in this study in comparison to available literature data was probably the result of a shorter grain conditioning time and lower temperature. Additionally, in a recent study by Kang, Lee, Jeon, Lee, Lee and Seo [22], seven steam-flaked corn samples were collected from multiple feed manufacturers. The range of DG was from 32 to 89%; however, the authors did not provide any information regarding the processing parameters.

In sample MC, which was conditioned to 14.36% moisture and micronized at a temperature of 111°C, the determined DG was 51.74%. In a study by Medel et al. [43], prior to micronization, corn grain was soaked for 24h until a moisture content of 17.5% was reached. In the above study, the maximum temperature of 70.7 °C was measured in the last part of the micronizer, which resulted in a DG of only 27%. The significantly higher DG obtained in this study for micronized corn was probably due to the significantly higher temperature in the last segment of the micronizer, despite the lower moisture content after conditioning.

A moderate DG of 35.64% was recorded in sample PC at a pelleting temperature of 61 °C. In a study by Moritz et al. [44], the DG of pelleted corn was slightly lower and was 29%. Svihus and Zimonja [45] reported that low DG after pelleting, which most often ranges from 5 to 30%, is due to the limited moisture content and moderate temperature during this heat treatment.

In contrast with the above-examined heat treatments that proved to be mild regarding DG, extrusion proved to be the most effective in breaking the structure of the starch granule. Only after this treatment, complete starch gelatinization (100%) for sample EC was achieved. During extrusion, DG depends to a great extent on the given conditions, and according to Svihus et al. [46], it is possible to achieve a gelatinization of only 40% of the starch by adjusting this treatment condition. In a study by Han, Guo, Cai, and Yang [24] and another by Moritz, Parsons, Buchanan, Calvalcanti, Cramer, and Beyer [44], the DG of extruded corn was 92%, while Medel, Salado, De Blas, and Mateos [43] recorded a slightly lower value of 76%. The complete gelatinization of the starch in sample EC is probably due to the larger shear forces and longer retention time of the material inside the extruder barrel under the given extrusion conditions [47].

### 4.2. Water Absorption Index (WAI) and Water Solubility Index (WSI)

WAI is a physicochemical parameter that measures the amount of water absorbed by starch and can be used as an index of gelatinization [35]. It has been reported that WAI indicates the hydrolytic breakdown of starch during heat treatments and the swelling behavior of the starch component [48]. The highest WAI was recorded for the extruded sample, in which the DG was 100%, based on DSC analysis. Correlation analysis confirmed a positive correlation between WAI and DG (r = 0.951, *p* ≤ 0.05).

According to White et al. [49], WSI indicates the amount of soluble polysaccharides released from the starch granules to the aqueous phase. However, Dogan and Karwe [50] state that WSI includes the opposing effects, i.e., starch dextrinization and interactions between degraded components at the molecular level. Steam flaking produced the lowest WSI (3.37 g/100 g), which was almost two times lower than in the unprocessed sample (6.20 g/100 g). Based on DSC analysis, the DG in SFC was 42.51%, and a decrease in WSI indicates a possible interaction between degraded components (starch, proteins and lipids) at the molecular level which in turn leads to an increase in molecular weight and decreased solubility, thus a decreased WSI [51]. Based on a slight decrease in WSI after pelleting (6.11 g/100 g) and micronization (5.91 g/100 g) in comparison to the unprocessed sample (6.20 g/100 g), it can be assumed that interactions on molecular level also occurred, but to a lesser extent than after steam flaking. On the other hand, the increase in WSI was observed only after extrusion. According to the starch degradation model during extrusion proposed by Gomez and Aguilera [52], three pure states (raw, gelatinized, and dextrinized) of starch exist together. According to the same model, dextrinization can be considered to take place along with or immediately after adequate gelatinization, and it leads to an increase in WSI. Based on a major increase in WSI after extrusion at the described parameters (more than three times that of the unprocessed sample) it can be assumed that complete gelatinization of starch (as confirmed by DSC analysis) was followed by dextrinization. Correlation analysis revealed that WAI and WSI are positively correlated (r = 0.936, *p* ≤ 0.05). 

In the available literature, WAI and WSI are used to evaluate the physicochemical changes that occur after heat treatments, most commonly after extrusion. An increase in WAI and WSI has been reported for samples of extruded corn [35,53], sorghum [54], barley [48], amaranthus [55], quinoa [50], and isolated starches (corn, potato, and tapioca starch) [36,56]. In a study by Deepa and Hebbar [57], an increase in WAI and a decrease in WSI was reported after the applied micronization treatment of corn grain, which is in agreement with the present results.

### 4.3. Pasting Properties

Liu [58] states that the paste viscosity cannot be produced by fragmented granules or solubilized starch substance, but only by intact swollen granules. Based on the above, the absence of a characteristic peak in sample EC confirms the complete gelatinization of the starch. Additionally, the extruded sample had a significantly higher IV when compared to unprocessed corn (19.90 vs. 1.07 Pa·s, *p* ≤ 0.05). The observed increase in IV at low temperature is due to the higher presence of soluble substances that ensued during heat degradation of starch, such as dextrin fragments, which is consistent with the results obtained for WSI. The results obtained for the extrusion treatment have been confirmed for corn [53,59], corn starch [56], amaranthus [55], and sorghum [54]. In addition to the extruded sample, the results for IV of the other heat-treated samples are also in accordance with the WSI, and thus for samples PC, SFK, and MC where a decrease in WSI was observed and IV also decreased. Correlation analysis confirmed a strong positive correlation between IV and WSI (r = 0.985, *p* ≤ 0.01). 

The decrease in viscosity while the sample is maintained at high temperature (in this case at 95 °C) is due to the breakdown of the starch granules followed by a linear orientation of the polymer, which reduces the viscosity of the paste. This was noted for the unprocessed sample (UC), as well as for the corn sample that was pelleted (PC). In contrast, in the SFC and MC samples, an increase in viscosity was found when the sample was maintained at high temperature, for which Gidley et al. [60] and Mahasukhonthachat et al. [54] state that it is associated with limited and incomplete starch swelling during the heating stage due to the connections that exist between starch and protein. The obtained results support the assumptions based on WSI analysis, i.e., after steam flaking and micronization there was an interaction between degraded components, namely starch and protein. Additionally, White et al. [33] state that there is possibly mobility of the amylopectin chains within the starch granules during micronization, leading to a greater rigidity of amorphous regions which could affect the swelling characteristics of starch granules and therefore the pasting curve.

For the unprocessed corn sample, the highest value of PV (68.06 Pa·s) was observed, as well as the highest value of T_g_ (77.72 °C). After the applied heat treatments, there was a decrease in PV value (at 95 °C) in all analyzed samples as compared to the unprocessed sample, which is due to the partial gelatinization of the starch during the heat treatment, i.e., a decrease in the number of granules whose swelling results in an increase in the viscosity of the paste. Correlation analysis confirmed a negative correlation between PV and DG (r = −0.967, *p* ≤ 0.01), as well as between PV and WAI (r = −0.955, *p* ≤ 0.05). A decrease in PV was observed in eight varieties of steam-flaked sorghum when compared to unprocessed in a study by Smith [61] using RVA. The same finding was reported using a Brabender amylograph in the study of McNeill et al. [62] for steam-flaked and micronized sorghum, and by Deepa and Hebbar [57] for two varieties of micronized corn. Additionally, after heat treatment and damage to the starch granules, they became less resistant to swelling, which was found on the basis of a decrease in T_g_ [63]. By analyzing the obtained pasting curves, in addition to reducing T_g_, one can also observe a decrease in the slope of the curve from the moment of viscosity increase to the maximum viscosity. The observed decrease in slope in all heat-treated samples is due to inhomogeneity, i.e., a different degree of damage to the individual granules after the applied treatment. In the UC sample, the viscosity increased in the shortest time interval and at the highest temperature due to the native starch granules, which are intact and more resistant to swelling. Correlation analysis indicated that T_g_ was positively correlated with PV (r = 0.887, *p* ≤ 0.05), and negatively correlated with WAI (r = −0.978, *p* ≤ 0.01), WSI (r = −0.982, *p* ≤ 0.01), and IV (r = −0.996, *p* ≤ 0.01).

In all analyzed samples, there was an increase in final viscosity during the cooling phase for which Singh et al. [37] state that it exhibits a tendency of the various constituents present in the hot paste (swollen granules, swollen granule parts, colloidally- and molecularly-dispersed starch molecules) to interact with each other when the paste temperature decreases, representing a retrogradation process. In extruded sample EC, it was determined that starch dextrinization has occurred in addition to gelatinization, based on WAI and WSI analysis, and the lowest value of FV was observed as a result of the presence of dextrins, which otherwise have low viscosity.

### 4.4. Scanning Electron Microscopy (SEM)

The observed changes in the structure are in accordance with the results of Harbers [64], who observed the rupture of sorghum starch granules and the formation of a shapeless mass after steam flaking using SEM. Wang and Copeland [39] state that depending on processing conditions, starch samples that are not 100% gelatinized may contain granules that are more similar to the native state (zero gelatinization), granules at different stages of partial gelatinization, as well as fully gelatinized granules. Based on SEM images, Bdour et al. [65] confirmed that barley and sorghum starch granules retain their integrity after pelleting, and the same is not the case after extrusion.

## 5. Conclusions

The application of an in vitro methodology led to important findings regarding the physicochemical changes in corn starch upon the application of heat treatments commonly used in animal nutrition (pelleting, steam flaking, and extrusion), as well as micronization, for which there is very little data in the literature. Based on the DG results, pelleting, steam flaking, and micronization can be considered to be mild heat treatments, whereas extrusion proved to be a severe heat treatment. Analysis of the functional and pasting properties implied a possible interaction between degraded components (starch, proteins, and lipids) in the steam-flaked sample, as well as in the micronized sample, though to a lesser extent. Additionally, the occurrence of dextrins was noted after extrusion under the given processing conditions. Various processing treatments of corn undoubtedly influence rumen fermentation characteristics; therefore, future research should be aimed at the determination of starch rumen fermentation characteristics and the possible correlations between applied analyses (used to quantitatively represent the physicochemical properties of processed starch) and rumen degradation traits.

## Figures and Tables

**Figure 1 animals-12-02234-f001:**
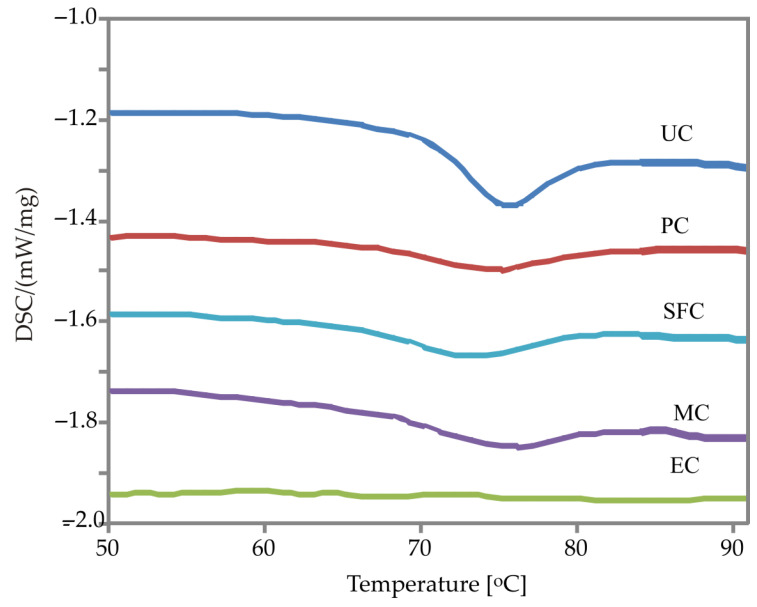
DSC thermograms of unprocessed and heat-treated corn samples (UC—unprocessed corn; PC—pelleted corn; SFC—steam-flaked corn; MC—micronized corn; EC—extruded corn).

**Figure 2 animals-12-02234-f002:**
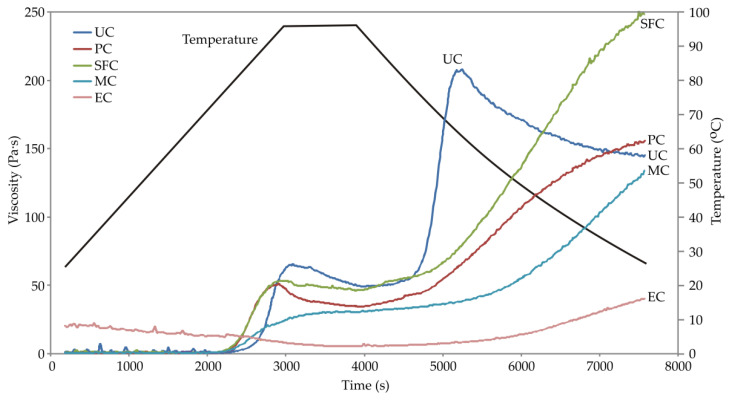
Pasting profiles of unprocessed and heat-treated corn samples (UC—unprocessed corn; PC—pelleted corn; SFC—steam-flaked corn; MC—micronized corn; EC—extruded corn).

**Figure 3 animals-12-02234-f003:**
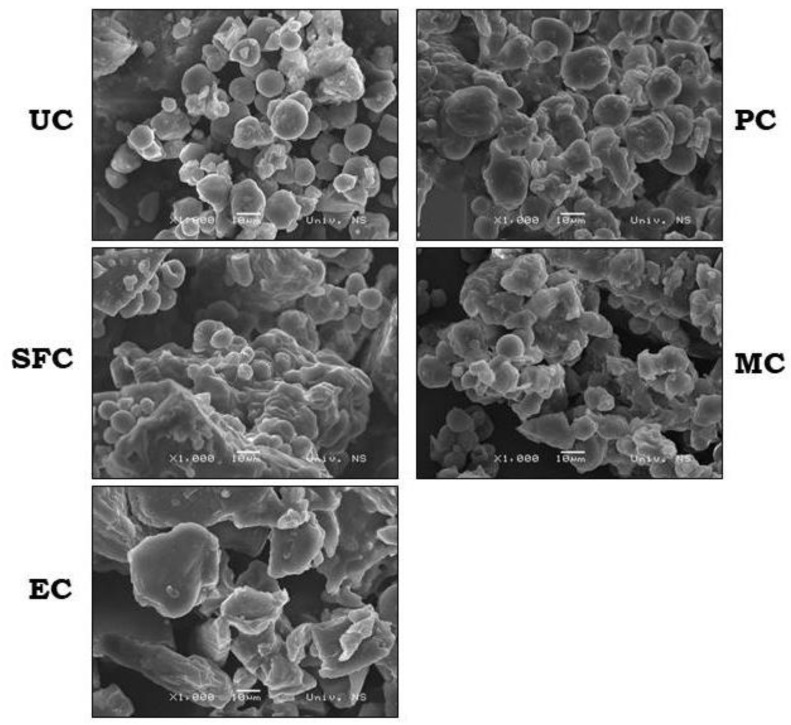
Scanning electron microscopy images of unprocessed and heat-treated corn samples (UC—unprocessed corn; PC—pelleted corn; SFC—steam-flaked corn; MC—micronized corn; EC—extruded corn).

**Figure 4 animals-12-02234-f004:**
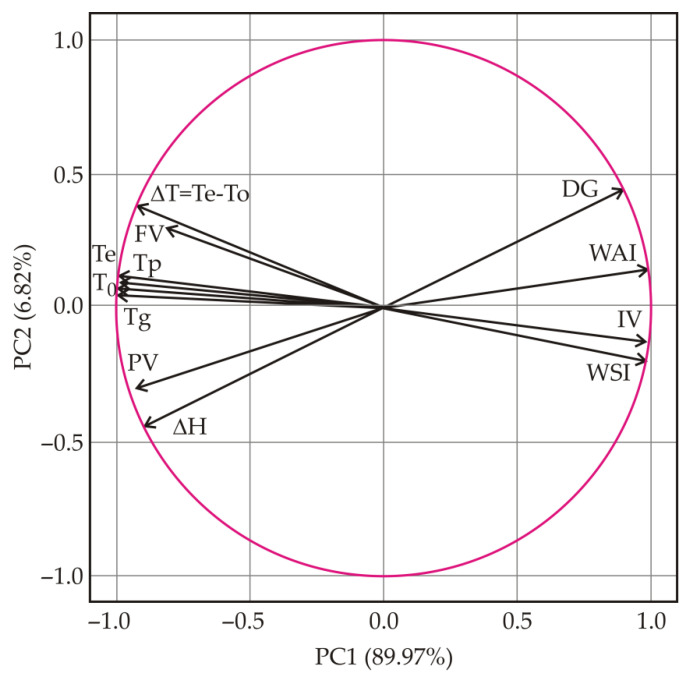
Principal component loading plot for unprocessed and heat-treated corn (onset temperature (T_o_), peak temperature (T_p_), end temperature (T_e_), and enthalpy change (∆H) of starch gelatinization; ∆T—gelatinization temperature range; DG—degree of gelatinization; IV—initial viscosity, PV—peak viscosity, FV—final viscosity, T_g_—gelatinization temperature; WAI—water absorption index; WSI—water solubility index).

**Figure 5 animals-12-02234-f005:**
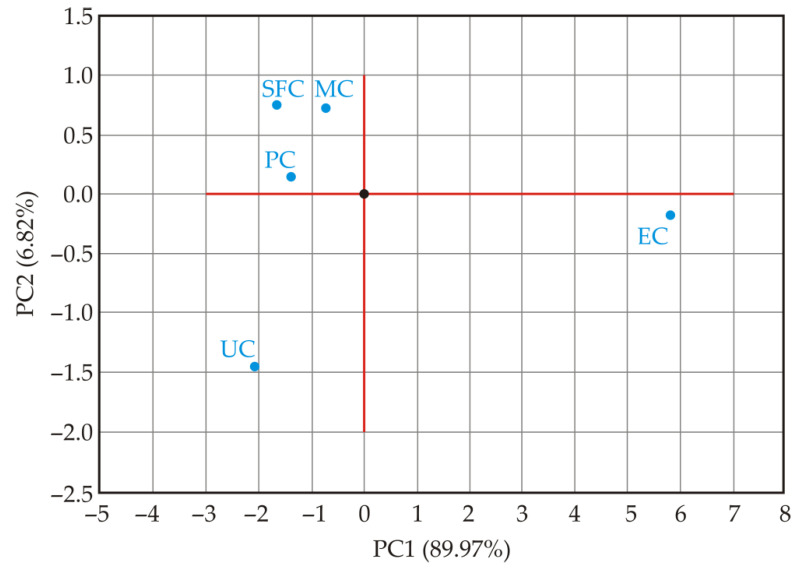
Principal component score plot of unprocessed and heat-treated corn (UC—unprocessed corn; PC—pelleted corn; SFC-steam-flaked corn; MC—micronized corn; EC—extruded corn).

**Table 1 animals-12-02234-t001:** Gelatinization, functional, and pasting properties of unprocessed and heat-treated corn.

Properties	UC	PC	SFC	MC	EC
Gelatinization					
T_o_ (°C)	71.0 ± 0.1 ^c^	67.1 ± 0.2 ^a^	66.9 ± 0.2 ^a^	68.7 ± 0.1 ^b^	/†
T_p_ (°C)	75.6 ± 0.2 ^c^	75.0 ± 0.3 ^b^	73.4 ± 0.2 ^a^	75.7 ± 0.1 ^c^	/
T_e_ (°C)	79.4 ± 0.2 ^b^	79.4 ± 0.3 ^b^	78.8 ± 0.2 ^a^	80.3 ± 0.2 ^c^	/
∆T = T_e_ − T_o_ (°C)	8.4 ± 0.1 ^a^	12.3 ± 0.2 ^c^	11.9 ± 0.3 ^bc^	11.6 ± 0.1 ^b^	/
∆H (J/g)	4.30 ± 0.09 ^d^	2.77 ± 0.07 ^c^	2.47 ± 0.05 ^b^	2.08 ± 0.10 ^a^	/
DG (%)	0.00 ^a^	35.64 ± 1.17 ^b^	42.51 ± 1.90 ^c^	51.74 ± 2.22 ^d^	100^e^
**Functional (hydration)**					
WAI (g/g)	2.77 ± 0.11 ^a^	3.35 ± 0.08 ^b^	3.55 ± 0.09 ^bc^	3.86 ± 0.10 ^c^	6.59 ± 0.20 ^d^
WSI (g/100 g)	6.20 ± 0.19 ^b^	6.11 ± 0.30 ^b^	3.37 ± 0.42 ^a^	5.91 ± 0.12 ^b^	20.89 ± 0.91 ^c^
**Pasting**					
IV (Pa·s)	1.07 ± 0.09 ^c^	0.42 ± 0.01 ^a^	0.77 ± 0.02 ^b^	0.49 ± 0.04 ^ab^	19.90 ± 0.24 ^d^
PV (Pa·s)	68.06 ± 2.85 ^c^	50.03 ± 0.07 ^b^	53.46 ± 0.22 ^b^	30.41 ± 0.08 ^a^	/
FV (Pa·s)	149.80 ± 5.00 ^c^	157.29 ± 1.70 ^d^	248.60 ± 0.70 ^e^	130.55 ± 3.05 ^b^	39.61 ± 0.74 ^a^
T_g_ (°C)	77.72 ± 0.02 ^d^	72.83 ± 0.31 ^b^	73.57 ± 0.01 ^c^	72.14 ± 0.29 ^a^	/

† no endotherm was recorded due to the complete gelatinization of starch. Values are means ± standard deviation of three replicates. Superscripts with different letters in same row indicate significant differences in means (*p* ≤ 0.05). UC—unprocessed corn; PC—pelleted corn; SFC—steam-flaked corn; MC—micronized corn; EC—extruded corn; onset temperature (T_o_), peak temperature (T_p_), end temperature (T_e_), and enthalpy change (∆H) of starch gelatinization; ∆T—gelatinization temperature range; DG—degree of gelatinization; IV—initial viscosity, PV—peak viscosity, FV—final viscosity, T_g_—gelatinization temperature.

## Data Availability

Not applicable.

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
