# Peer review of "Physicochemical Changes of Heat-Treated Corn Grain Used in Ruminant Nutrition"

_animals, 2022, doi:10.3390/ani12172234_

Round 1

Reviewer 1 Report

  • The proposed article for review titled “Physicochemical changes of heat-treated corn grain used in ruminant nutrition” has a well-defined objective – to evaluate effects of different heat treatments (pelleting, steam flaking, micronization and extrusion) on the physicochemical properties of raw and processed corn starch using in vitro methodology for the of the ruminant nutrition purposes. The authors state the objective of the article after a well-written introduction, where the gap that clearly exists in animal science literature, as well as the importance of efforts to gain basic knowledge of the physicochemical properties of heat treatments corn starch, are very well argued. The objective is also very ambitious, due to the facts that the feed technology is an important scientific field that is related to and precedes animal nutrition and has an important contribution to utilization of nutrients. Animal nutritionists pay a special attention to starch from cereals as non-structural carbohydrates, because meeting the energy needs of animals depends largely on its degradability properties.

  • Criticism

    Overall, the article undoubtedly has a high scientific value due to the evaluated effects of heat treatments of corn starch. A wealth of scientific information was obtained on a large set of measured variables. The discussion made is relevant. However, can the question of the article suitability for the journal be answered in the affirmative? On this issue the position of the authors should be reconsidered. For example, given the scope of the scientific journal, it would be better to show in the discussion which changes in the measured parameters are extremely good for the animals, which are not good enough. 

    A large number (64 in total) of literary sources are cited, but only 12.5% are from the last five years, which does not meet the requirements. In such a case, it would be better to specify both the number and the references cited.

    Regarding the cited references, three characteristics were noted as weaknesses:

    * On l. 67 (p.2) has a citation [3, 6-24] that I think needs correction, so many references in one place are not acceptable.

    * Analysis of writing style, in particular in the Introduction and Discussion, showed that when the cited references are numbered (rather than with the author`s name), then the sentences should be worded differently. It doesn`t sound good: "According to [3]" (l.54) as well as l.87, l. 375, l. 387 etc; "In the study of [43]", l.367; "... for which [59] and [53] state...", l. 433 etc.

    * Starting the sentence with number also does not sound good - l.344, l.350, l. 368, l. 415 etc.

    All of the above indicates that due to citation with the source number both sections shown (Introduction and Discussion) would benefit from a careful editing.

                  The paragraph numbers on l. 177, l. 189 and l.195 to be corrected from 2.4 to 2.5;                     from 2.5 to 2.6 and from 2.6 to 2.7 respectively.

Author Response

Dear reviewer,

All authors of the submitted manuscript would like to thank you for your time and effort to make this manuscript better.

Please find below our answers in the attachment. 

Reviewer 2 Report

The article addresses an important topic in the field of feed technology and processing. in line with this hover they should consider also in the introduction that some nutritional evaluation have been proposed recently such us the glycemic index potential as report by    Ottoboni et al  DOI: 10.1080/1828051X.2019.1596758  which is important for assess such type of material for feed. this should be mentioned at list in the conclusion since the proposed evolutions are Emily analitica and can be completed by a digestibility and function evaluation like Glycemic index potential and HI. 

Again other recent article have reported quite similar effect on starch and other sugar fraction in feeding stuffs like former food (containing gelatinized starch).  for instance

1.     Pinotti et al..DOI: 10.1016/j.jclepro.2021.126290 

2.     Luciano et al DOI: 10.3390/ani10010125 

these works are missing in the introduction e.g. in line 69 it can be added that new and emerging feed material can contain and condense such feautures like former foodstuffs.  

line 415 are you sure to start with [57]?

Author Response

(The authors gave the same response as above.)
